# Assessment and Optimization of Thermal Stability and Water Absorption of Loading Snail Shell Nanoparticles and Sugarcane Bagasse Cellulose Fibers on Polylactic Acid Bioplastic Films

**DOI:** 10.3390/polym15061557

**Published:** 2023-03-21

**Authors:** Oluwatoyin J. Gbadeyan, Linda Z. Linganiso, Nirmala Deenadayalu

**Affiliations:** 1Green Engineering Research Focus Area, Faculty of Engineering and Built Environment, Durban University of Technology, Durban 4001, South Africa; 2Department of Chemistry, Durban University of Technology, Durban 4001, South Africa

**Keywords:** bioplastic, sugarcane bagasse cellulose fibers (SBCF), snail shell nanoparticles (SSNP), polylactic acid (PLA), thermal stability and water absorption

## Abstract

The optimization and modeling of the parameters, the concentration of polylactic acid (PLA), sugarcane bagasse cellulose fibers (SBCF), and snail shell nanoparticles (SSNP), were investigated for the development of bioplastic films. With the aid of the Box–Behnken experimental design, response surface methodology was used to assess the consequence of the parameters on the water absorption and thermal stability of fabricated bioplastic films. Varied water absorption and thermal stability with different component loading were obtained, evidencing the loading effect of snail shell nanoparticles and sugar bagasse cellulose fibers on bioplastic film’s water absorption and thermal stability. The quadratic polynomial model experiment data offered a coefficient of determination (R^2^) of 0.8422 for water absorption and 0.8318 for thermal stability, verifying the models’ fitness to develop optimal concentration. The predicted optimal parameters were polylactic acid (99.815%), sugarcane bagasse cellulose fibers (0.036%), and snail shell nanoparticles (0.634%). The bioplastic developed with optimized concentrations of each component exhibited water absorption and thermal stability of 0.45% and 259.7 °C, respectively. The FTIR curves of bioplastic films show oxygen stretching in-plane carbon and single-bonded hydroxyl bending in the carboxylic acids functional group. SEM and TEM images of the bioplastic showed dispersion of the nanoparticles in the matrix, where SSNP is more visible than SBCF, which may be due to the lesser loading of SBCF. The improved properties suggest an optimum concentration of naturally sourced resources for developing bioplastic, which may be used for food and drug packaging for delivery.

## 1. Introduction 

Plastics have become part of our daily life as they are widely used for several applications [1]. Plastic usage in different industries, including electrical and electronic, telecommunications, clothing, and food industries, increased the amount of waste plastic in our environment and water [1]. Besides, plastic, being lightweight, durable, and low cost, has increased been subject to increased demand for packaging, especially in the medical, agricultural, and food industries [2]. The increasing demand for plastic consequently increases its waste. According to United Nations Environment Programme (UNEP), the amount of plastic generated globally in 2018 was roughly 360 million tons and is forecast to increase at an annual compound growth rate of 4.2% between 2021 and 2028 [3]. Most plastic in circulation, either thermoplastic or thermosetting, is chemically produced. Plastics are widely deployed in several applications due to their availability, durability, and easy processability [4]. Despite the robustness and durability of these chemically produced plastics, they contain toxic elements dumped into the environment and harm the ecosystem [4,5,6]. They also present ecological hazards to the environment and human health due to their resistance to natural forces.

Consequently, plastic waste is often controlled by dumping it into landfills, and some countries choose to burn it. However, larger land spaces that could be used for farming are used for this purpose, threatening the ecosystem’s stability [7]. Similarly, burning petroleum-based plastics emits CO_2_ into the atmosphere, threatening human and aquatic life [8,9,10]. Several sensitizations on these concerns are ongoing to accelerate the reduction of petrochemical-sourced plastic [11,12,13]. Besides, the continually used of these materials may not alleviate the havoc caused by plastic materials on the ecosystem because these materials contain some toxic elements. Therefore, a combination of biodegradable materials is needed to develop bioplastic to get a lasting solution to the damage wrought by waste plastic on the environment, which means transferring from petrochemical-sourced plastic to naturally sourced bioplastic. Given this, researchers have developed different naturally resourced bioplastics and films to serve as alternative plastic materials. Several bioplastics have been developed using chemically processed, biobased polymers, such as poly (vinyl alcohol) (PVA), polylactic acid (PLA), and polycarbonate (PC) with effective biodegradable content [3,4]. These materials have become a choice of material for developing bioplastic films [14,15,16]. However, their drawbacks include inadequate strength and stiffness with poor thermal stress resistance, which opens research opportunities for other suitable additives to improve the properties of the plastics. 

Several organic and inorganic reinforcements have been incorporated to improve bioplastic properties, which may be single or hybrid fillers. The enhancement of the bioplastic’s properties is aimed at increasing the suitability of bioplastic as a potential alternative to petrochemical-sourced plastic. Several filters have been employed to produce bioplastic with improved thermomechanical properties [17,18,19,20]. Factors, such as particle sizes and incorporated reinforcement, often significantly affect bioplastic production and properties. Consequently, researchers incorporated single filler to improve bioplastic thermomechanical and physical properties and [3,17,18,19,20,21,22,23,24]. Despite numerous studies on improving bioplastic films by incorporating reinforcements, there are limited studies where polylactic acid reinforced with sugarcane bagasse cellulose fibers (SBCF) hybridized with snail shell nanoparticles (SSNP) are reported.

Our previous study showed the reinforcement effect of snail shell nanoparticles (SSNP) on PLA thermal stability at low loading [4,25,26,27]. Incorporating sugarcane bagasse cellulose fibers (SBCF) has been shown to potentially improve thermal stability and rheological and optical properties [28,29,30,31,32]. Therefore, combining these fillers into PLA is anticipated to produce bioplastic films with enhanced properties. Nevertheless, the fillers loading ratio often has substantial effects on the properties of bioplastic material, which may be either negative or positive. However, limited literature reports the loading optimization of additives to produce bioplastic with improved properties. Therefore, investigation towards optimizing the loading percentages of fillers to develop bioplastic with improved properties is needed. Optimization of sugarcane bagasse cellulose fibers and snail shell nanoparticles concentration on the water absorption and thermal stability of PLA-based bioplastic films is the primary aim of this study. The effect of these loading parameters was investigated using a response surface methodology (RSM) design of experiments. 

## 2. Material and Methods

### 2.1. Materials

Polylactic acid (PLA bioplastic pellets) used as a binder was supplied by Sigma-Aldrich (USA). Acetone with 99.90% purity adopted for liquifying PLA bioplastic pellets was purchased from Laboratory and Analytical supplies in Durban, South Africa. Snail shell nanoparticles and sugarcane bagasse cellulose fibers produced through extraction were used as reinforcement. The snail shell nanoparticles containing 100% calcium carbonate with 100% purity and aragonite polymorph structures shown in Figure 1 were produced using mechanochemical techniques (ball milling) at the chemical department of the Durban University of Technology, South Africa [26].

#### 2.1.1. Techniques for Cellulose Fibers Extraction from Sugarcane Bagasse

Cellulose fibers were extracted from sugarcane bagasse by milling into smaller sizes ranging from 1 to 5 mm. These cellulose fibers were extracted using alkali treatment, bleaching, and acid hydrolysis. The chopped fiber was treated with alkali chemicals using a 6% sodium hydroxide (NaOH)/distilled water solution. Forty grams of sodium hydroxide (NaOH) were dissolved in 800mL of distilled water. Then, 70 g of fibers were drenched in sodium hydroxide (NaOH) solution for 30 min. This process removes lignin, hemicellulose, wax, and oils from the fiber. Then, the bleaching process followed and completed in three stages. Before bleaching, the fiber was dehydrated in an oven at 60 °C for 4 h. Then, the dehydrated fibers were drenched in concentrated sodium hypochlorite (NaOCl) solution for 30 min. This process was repeated thrice. The fibers were rinsed in distilled water to get rid of residue chemicals and dehydrated at 60 °C for 4 h. This process was repeated twice to ensure purification. The fiber was then hydrolyzed with acid [35] in a sulfuric acid solution. The sulfuric acid and distilled water were diluted using a 1.10 mixing ratio. After that, impurities and excess sulfuric acid in the fiber were eliminated by rinsing the fiber in running distilled water to eliminate impurities and excess sulfuric acid in the fiber. The fibers were placed in an oven and dried at 60 °C for 4 h, and then ground and sieved to obtain 16 g of cellulose nanoparticle fibers with an average size of 0.75 microns.

#### 2.1.2. Experimental Design and Optimization of Bioplastic Film Production

The Box–Behnken method of design expert software was used to design the number of trials runs for optimizing bioplastic film production. Three independent variables in the range of 99–100 wt.% for PLA, 0.1 wt.% for SBCF, and 0.5–1 wt.% for SSNP were used, and water absorption and thermal stability were considered as the dependent variables. In this regard, the software above generated seventeen investigational runs, as shown in Table 1.

The investigational data were inputted into the polynomial quadratic model equation to narrate the independent and the dependable variables, in agreement with Equation (1).
Y = Ƌ_0_ + Ƌ_1_X_1_ + Ƌ _2_X_2_ + Ƌ _3_X_3_ + Ƌ _11_X_1_X_1_ + Ƌ_12_X_1_X_2_ + Ƌ _13_X_1_X_4_ + ժ_11_X_1_^2^ + ժ_22_X_2_^2^ + ժ_33_X_3_(1)
where Y signified the response outputs and Ƌ_0_ the intercept; the linear coefficient was obtained from Ƌ _1_X_1_ to Ƌ _4_X_4_; Ƌ_12_X_1_X_2_ to Ƌ_23_X_2_X_4_ represents the interactive coefficients, and Ƌ_11_X_1_^2^ to Ƌ_33_X_3_^2^ the quadratic coefficients. The consequence validation of the model was assessed by evaluating variance (ANOVA) with a 95% confidence interval (CI band). Three-dimensional response surface plots were fashioned to evaluate the interaction of variable parameters.

#### 2.1.3. Bioplastic Preparation

PLA-based bioplastic reinforced with snail shell nanoparticles and sugarcane bagasse cellulose fibers was prepared via solvent casting. This fabrication process was conducted in four stages. The cellulose fibers from sugarcane bagasse extraction mentioned above were the first stage. The second step was drying PLA bioplastic pellets to ensure the granules’ hydrophobicity and the PLA’s liquifying in solvent (acetone). The dispersion of the snail shell nanoparticles and sugarcane bagasse cellulose fibers in the PLA/acetone solution, which is the third step, followed. The fourth step involved forming bioplastic films in a mold. In the first step, 100 g of PLA bioplastic pellets were measured and dried at 60 °C in a vacuum oven for 30 min to remove moisture. PLA bioplastic pellets were liquified in acetone using a 1:20 liquifying ratio. The PLA bioplastic pellet and acetone portion were measured into a container and covered for 72 h with a lid. Afterward, the PLA/acetone blend was mechanically stirred at 500 rpm to ensure the complete suspension of PLA. The PLA bioplastic pellets blend was allowed to settle to avert bubbles that caused porosity in plastic material. Then, sugarcane bagasse cellulose fibers and snail shell nanoparticles were slowly added to PLA/acetone blend according to the design presented in Table 1 and stirred further at 500 rpm for 15 min to facilitate uniform dispersion of the PVA, sugarcane bagasse cellulose fibers, and snail shell nanoparticles. Later, the blend was poured into an open-plastic mold coated with wax and cured for 72 h at ambient temperature. However, the bioplastic film shown in Figure 2 was characterized after two weeks to ensure maximum sample curing.

#### 2.1.4. Water Immersion

The water absorption of snail shell nanoparticles and sugarcane bagasse cellulose fibers reinforced polylactic acid was determined by immersing it in 100 mL of water for 24 h at ambient temperature. This investigation was conducted according to ASTM D570-98 standard test specifications. Before immersion, the bioplastic firm was dried in the oven at 60 °C for 2 h to dehydrate the sample. Then, the initial weight (T1) of five samples was taken for each bioplastic with different formulations. The samples were removed from the oven, immersed in water for 24 h at room temperature, wiped with a dry napkin, and weighed to determine the final weight of samples (T2) using a Pioneer digital electronic scale with 0.0001 g precision (Model PA214) manufactured in China. Water absorption % was measured using the equation below.
$$T_{B}=\frac{T_{2}-T_{1}}{T_{1}}\times 100$$
where *T_B_* = the percentage of water absorption, and the average *T_B_* value of the five samples is reported [25,35].

#### 2.1.5. Thermal Properties

Bioplastic thermal stability was determined on a thermal Analyser (Mettler Toledo TGA) with STARe software supplied by Microsep South Africa. The test was conducted under an inert atmosphere using a 100 mL/min dry nitrogen gas flow rate at 10 °C/min heating rate from 25 °C to 600 °C. 

#### 2.1.6. Fourier Transform Infrared Spectroscopic Analysis

The functional group of the bioplastic was determined on Fourier to transform infrared spectroscopy (FTIR) (Perkin Elmer, Waltham, MA, USA). A universal attenuated total reflectance module was used for the spectra of the analyzed samples in a wavenumber range between 4000 and 600 cm^−1^.

#### 2.1.7. Scanning Electron Microscopy (SEM) Analysis

The bioplastic microstructural properties were investigated to determine particle dispersion in the matrix and the mechanism governing the developed bioplastic properties. This microstructure investigation was conducted using ZAISS ultra FEG-SEM. Before the investigation, the bioplastic sample’s surface was gold-coated with an electronic thin gold.

#### 2.1.8. Transmission Electron Microscope (TEM)

The dispersion of snail shells nanoparticles was observed under a Philips CM120 BioTWIN transmission electron microscope (TEM). The microscopic investigation was carried out on specimens prepared using a KB/Wallac Type 8801 Ultramicrotome with Ultratome III 8802A Control Unit. Ultra-thin transverse sections approximately 80–100 nm thick were cut at room temperature using a diamond knife and supported by 100 copper mesh grids sputter-coated with a 3-nm-thick carbon layer. 

## 3. Results and Discussion

### 3.1. Bionaoparticle Optimization for Bioplastic Films Development

Table 2 shows water absorption and thermal stability of snail shell nanoparticles and sugarcane bagasse cellulose fibers reinforced polylactic acid developed using varied loading of SSNP, SBCF, and PLA. The relationship between the bioplastic films’ water absorption, thermal stability, and different loadings were generated using second-order polynomial Equations (2) and (3).
Water absorption (%) = +1.36 + 0.0386A + 0.3691B − 0.5983C − 0.0523AB − 0.1481AC − 0.05245BC + 0.3611A^2^ + 0.5932B^2^ − 0.1496C^2^(2)
Thermal stability (℃) = +247.90 − 4.75A + 6.27B + 9.52C + 19.58AB − 10.17AC − 2.88BC + 26.11A^2^ − 12.76B^2^ + 18.90C^2^(3)
where A, B, and C denote the concentration of SSNP, SBCF, and PLA in weight percentage. 

ANOVA was used to validate the tightfitting model authenticity, as shown in Table 3 and Table 4. High F-values of 4.15 and 3.85 with a low p-value of 0.0369 and 0.0447 of water absorption and thermal stability exhibited by bioplastic films are presented in Table 3. Whereas the coefficient of determination (R^2^) values obtained for these two models were 0.8422 (water absorption) and 0.8318 (thermal stability), indicating that the models could account for up to 84.2% and 83.2% of the dissimilarities in the investigational output.

### 3.2. The Impact and Interaction Effects of Parameters on Water Absorption and Thermal Stability

Figure 3 and Figure 4 present the impact and interaction effects of additive concentration at different loading parameters on the developed bioplastic firms’ water absorption and thermal stability. 

#### 3.2.1. The Interaction Effect of PLA and SBCF on Bioplastic Water Absorption and Thermal Stability

The interaction impact between PLA and SBCF loading on water absorption and thermal stability is shown in Figure 3a and Figure 4c. A significantly lower water absorption of 1.2027% was observed with a concentration loading of 100 wt.% PLA and 0.5 wt.% SBCF, which was below the predicted value. However, with the addition of 0.5 wt.% SSNP, lower water absorption of 1.3711% (at a combination of 99.5 wt.% PLA, 0.5 wt.% SBCF, and 0.5 wt.% SSNP) above design point predicted values were obtained. These combinations were identified as optimum based on the actual concentration factor of 0.5 wt.% SBCF. Snail shell nanoparticles increase the interfacial bonding of SBCF and PLA, resulting in solid structures with better barrier properties against water permeability [36,37,38]. A similar inclination was observed in Figure 2c for thermal stability, where a combination of 99.5 wt.% PLA, and 0.5 wt.% SBCF offered maximum thermal stability of 263 °C, indicating the optimal concentration of PLA and SBCF for higher thermo-physical properties of bioplastic films. These results were consistent with other findings reported elsewhere, where the loading of SBCF was found to improve thermal stability [39,40]. This significant improvement in water absorption rate may be related to the oxidation of the solvent fabricating bioplastic films [41]. The presence of acetone catalyzes against the bioplastic water absorption rate, resulting in the observation of low water permeability.

#### 3.2.2. The Interaction Effect of SBCF and SSNP on Bioplastic Water Absorption and Thermal Stability

Figure 3b and Figure 4a present the interaction effects of SBCF and SSNP, principal reinforcements on bioplastic films’ water absorption, and thermal stability. Significant low water absorption of 1.3711% was obtained after the combined 0.5 wt%. of SBCF and 0.5 wt% SSNP. This combination was acknowledged as optimal based on the concentration of 99.5% PLA. A notable interaction between SBCF and SSNP for thermal stability was observed at a combination of 1 wt%. SBCF and 1 wt%. SSNP exhibits superior thermal stability of 290.8 °C. This combination was identified as optimum based on the concentration of 99.5% PLA. The reinforcement effect of incorporating snail shell nanoparticles has been reported in previous studies, where the loading of SSNP positively improved the thermal stability of the developed bioplastic film [3,4]. This improved thermal stability may be attributed to inherent thermal properties and uniform dispersion of SSNP in the matrix of the films. Furthermore, this performance illustrated the reinforcement power of loading small nanoparticles. These two nanofillers are carbon-based, forming a synergistic effect, leading to catenation capability, increasing the chemical linkage into chains of atoms of the same element, and forming relatively strong bonds with itself and other components, resulting in better water resistance to permeability and improved thermal stability [3,42]. 

#### 3.2.3. The Interaction Effect of PLA and SSNP on Bioplastic on Bioplastic Water Absorption and Thermal Stability

The interaction of PLA and SSNP on the bioplastic film’s water absorption and thermal stability are shown in Figure 3c and Figure 4b. Outstanding low water absorption of 1.3711% was observed for small loading of 0.5 wt% SSNP, similar to what was observed after PLA and SBCF were combined. This optimal concentration was recognized based on the concentration of 0.5% SBCF. A similar trend was observed for bioplastic film’s thermal stability, as superior thermal stability is achieved when 99.5 wt.% PLA and 0.5 wt.% SSNP were combined. This trend emphasized the effectiveness of incorporating small quantities of snail shells nanoparticles. This performance may be attributed to improved reinforcement properties of SSNP. The hydrophobic nature of the SSNP is another reason for the decreased water-absorption rate observed. The improvement in thermal stability may be attributed to SSNP’s inherent thermal properties and homogeneous dispersion in the composite bioplastic film.

### 3.3. Justification of Water Absorption and Thermal Stability of Snail Shell Nanoparticles and Sugar Bagasse Cellulose Fibers Reinforced Polylactic Acid Bioplastic Films

The optimum setpoints associated with the model presented in Table 5 include polylactic acid (99.815%), sugar bagasse cellulose fibers (0.036%), snail shell nanoparticles, and (0.634%). The optimized process conditions for bioplastic films based on the model predicted water absorption and thermal stability of 1.42% and 240.248 °C, as shown in Table 6. The water absorption and thermal stability of bioplastic film developed with optimized additive concentration according to the model prediction were investigated at the same process conditions. The experimental process was repeated thrice to validate the accuracy of values obtained from the software, and the mean of three samples was reported. Bioplastic films with optimized component concentration exhibited 0.45% water absorption and 259.07 °C thermal stability, as presented in Table 6.

### 3.4. Characterization

#### 3.4.1. Fourier Transform Infrared Spectrometry

The functional group of the developed bioplastic films with optimum loading of SSNP, SBCF, and PLA was investigated using a Fourier transform infrared spectrum, as shown in Figure 5. The investigation was conducted considering wavelengths within the 550–4000 cm^−1^ range with numerous peaks. The medium spectrum peak at 2926 cm^−1^ was associated with C-H stretching vibration, presenting the distinctive feature of cellulose nanocrystals. It is also associated with the aldehyde band in the materials [3,27]. A protuberant peak observed at 1748 cm^−1^ is ascribed to strong C=O stretching from the δ-lactone compound, 1652 cm^−1^ allied with the C=O stretching from the δ-lactone compound class, and the C−O vibration peak at 1183 cm^−1^ is linked to the incorporation of PLA segments [43,44]. 

The absorption peak 1592 cm^−1^ due to N–H (–NH2 bending) of amine characterizes the absorption frequencies of β-D-pyranoside in snail shells nanoparticles [44]. Furthermore, the peak at 1452 cm^−1^ belongs to C-O stretching, and the absorption peak at 1362 cm^−1^ is due to the O-H bending of a primary alcohol, which may be associated with the PLA. FT 1080 cm^−1^, 1043 cm^−1^, 916.92 cm^−1^, and 870 cm^−1^ belonging to strong C-H bending were associated with carbonate ions connected with the snail shell nanoparticles in the film [25]. FTIR spectra displayed a dominant interface between C-O stretching, C=O stretching, and C-H stretching bending in the carboxylic acids, suggesting a bioplastic film functional group.

#### 3.4.2. Microstructure of Bioplastic Film

The developed bioplastic films with optimum loading of SSNP, SBCF, and PLA observed using scanning electron microscopy (SEM) at 38,000× magnification to determine the fracture mechanism, as presented in Figure 6.

An interlocking texture of the SSNP, SBCF, and PLA was observed. Although SSNP is visible on the surface of the bioplastic film investigated under SEM, the SBCF was not that visible, even at 3800× magnification. This feature may be attributed to the lesser loading of SBCF. Furthermore, the uniform dispersal of filler SSNP and SBCF decisively fused within the matrix, providing a reasonably flat surface that could be the reason for the barrier to a water absorption rate. The improved thermal stability may be attributed to visible SSNP on the developed bioplastic films. A microscopic investigation was conducted using a transmission electron microscope (TEM) to support the structural formation claims of the developed bioplastic with optimum concentration.

Figure 7 shows the TEM micrograph of the bioplastic sample. A consistent dispersion of sugarcane bagasse cellulose fibers (SBCF) and snail shell nanoparticles (SSNP) polylactic acid matrix was observed. This texture evidences the compatibility of the reinforcement filler and fiber and the mixing technique’s effectiveness. A uniform dispersion of reinforcement filler and fiber was observed without any agglomeration, proving the significant influence of the mixing techniques on the degree of dispersion observed. This structural packing formation without agglomeration may be attributed to optimum component concentration, forming a structure that resists water permeability.

Furthermore, calcium carbonate possesses higher thermal properties, and dispersing it at lower concentrations has been reported to influence polylactic acid-based bioplastic positively [3,4,26]. Similarly, calcium carbonate is hydrophilic, and its incorporation into bioplastic resulted in a low water absorption rate. In addition, components of the developed bioplastic are carbon-based additives, and a combination of these materials often forms catenation, resulting in a robust structural formation that resists water absorbency. These materials often transfer atoms from one to another to improve the properties of the final composite. The nano snail shell nanoparticle (calcium carbonate) may have transferred its hydrophilic and inherent thermal properties after being dispersed in bioplastic, leading to the improved thermal stability and water absorption of the bioplastic with optimal concentration. 

## 4. Conclusions

The fabrication process and additive concentration models of bioplastic films relating to the design effect of a combination of various loadings of Snail shell nanoparticles and sugar bagasse cellulose fibers’ water absorption and thermal stability were studied. Furthermore, the optimum concentration of Snail shell nanoparticles, sugar bagasse cellulose fibers, and polylactic acid for developing bioplastic with improved resistance to water uptake and enhanced thermal stability was designed and investigated using Box–Behnken experimental design. Three independent variables in the range of 99–100 wt.% for PLA, 01 wt.% for SBCF, and 0.5–1 wt.% for SSNP, were used with the water absorption and thermal stability selected as the dependent/response variables. The optimum setpoints associated with the model are polylactic acid (99.815%), sugar bagasse cellulose fibers (0.036%), and snail shell nanoparticles (0.634%). Bioplastic films with optimized component concentration exhibited 0.45% and 259.07 °C. Snail shell nanoparticles, sugar bagasse cellulose fibers, and polylactic acid loading positively influenced the predictable physio-thermal property setpoints. FTIR spectra displayed a dominant interface between C-O stretching, C=O stretching, and C-H stretching bending in the carboxylic acids, suggesting a bioplastic film functional group. SEM images reveal the homogeneous distribution of filler SSNP and SBCF firmly bonded within the matrix, resulting in a reasonably flat surface that may be described as a barrier for water absorption and improved thermal stability.

## Figures and Tables

**Figure 1 polymers-15-01557-f001:**
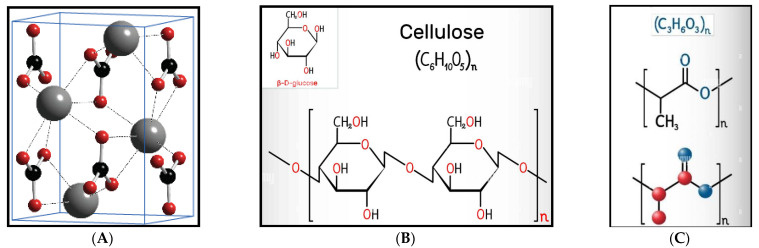
Image showing (**A**) calcium carbonate aragonite polymorph structures [33], (**B**) chemical structure of cellulose, and (**C**) polylactic acid [34].

**Figure 2 polymers-15-01557-f002:**
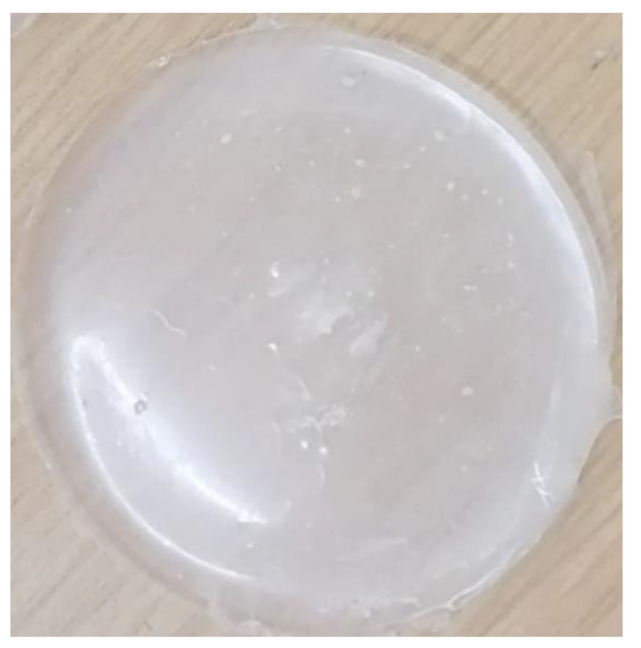
Snail shell nanoparticles and sugar bagasse cellulose fibers reinforced polylactic acid bioplastic film.

**Figure 3 polymers-15-01557-f003:**
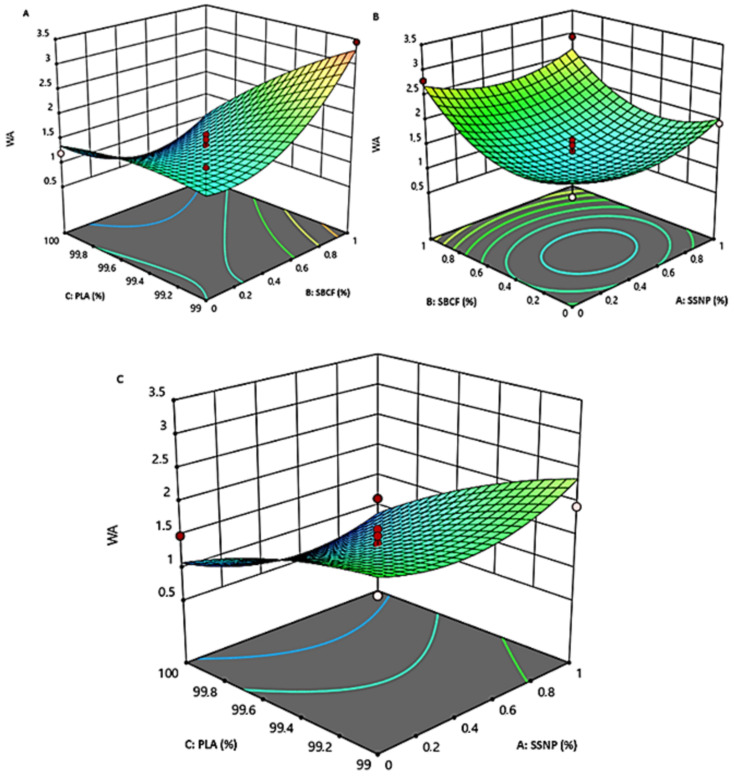
The 3-D surface plot of water absorption (WA) as a function of SSNP, SBCF, and PLA loadings: interaction effects of PLA and SBCF (**A**), SBCF and SSNP (**B**), PLA and SSNP (**C**).

**Figure 4 polymers-15-01557-f004:**
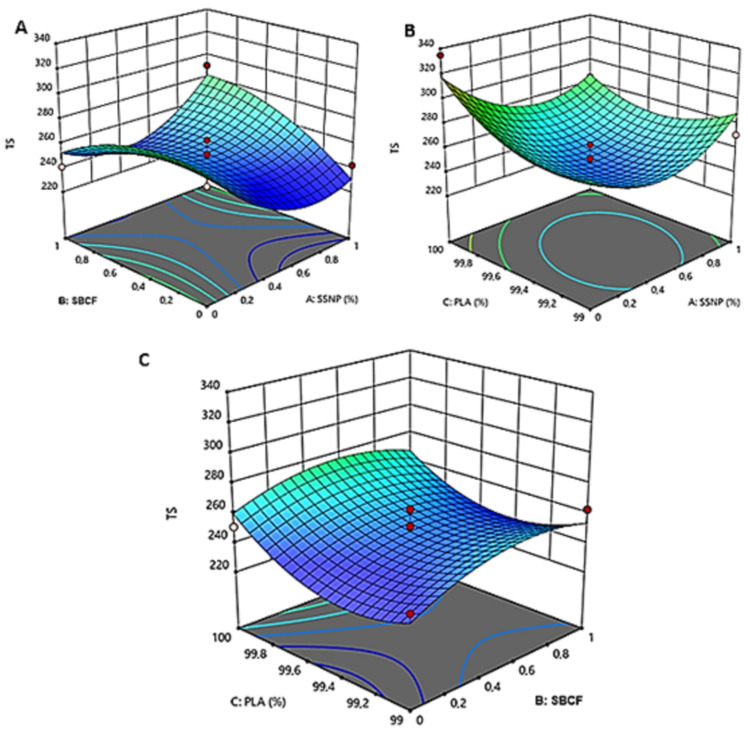
The 3-D surface plot of thermal stability (TS) as a function of SSNP, SBCF, and PLA loadings: interaction effects of PLA and SBCF (**A**), SBCF and SSNP (**B**), PLA and SSNP (**C**).

**Figure 5 polymers-15-01557-f005:**
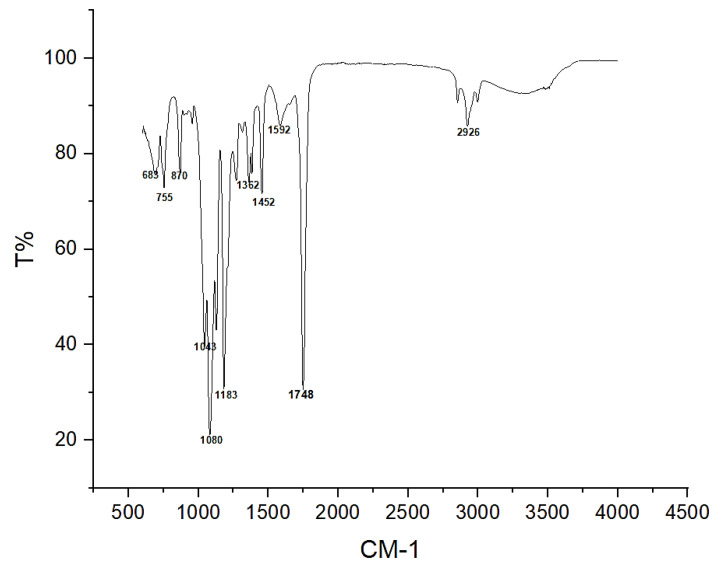
Graph showing the bioplastic film FTIR spectra.

**Figure 6 polymers-15-01557-f006:**
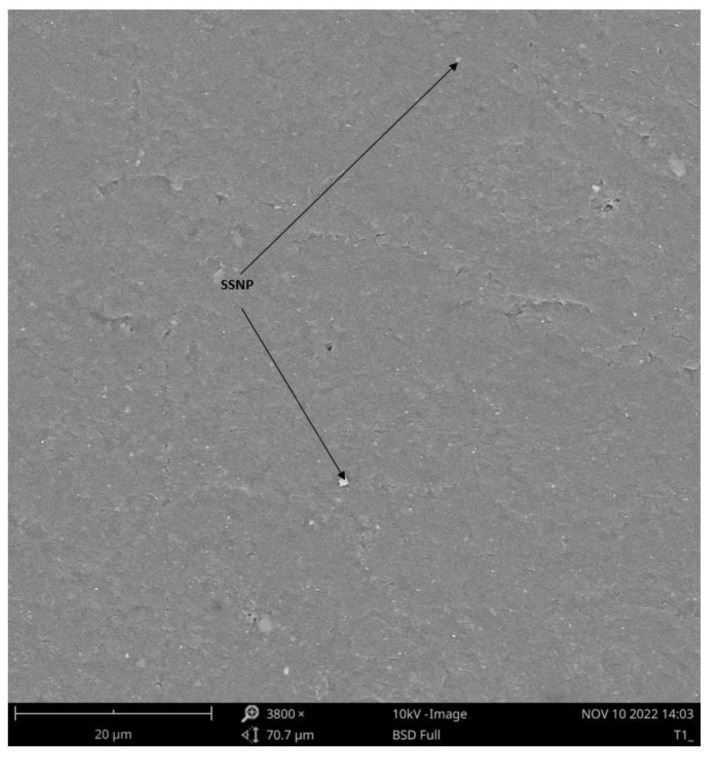
SEM image of the developed bioplastic film.

**Figure 7 polymers-15-01557-f007:**
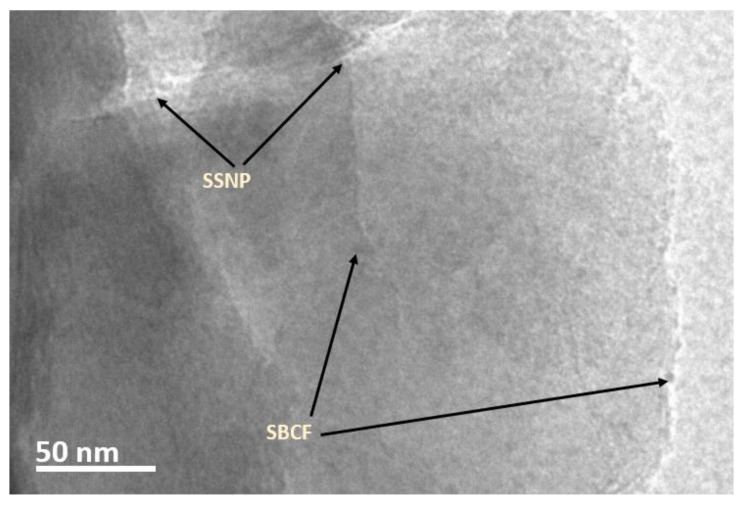
TEM micrograph showing the dispersion of sugarcane bagasse cellulose fibers and snail shell nanoparticles reinforced polylactic acid bioplastic.

**Table 1 polymers-15-01557-t001:** Investigational runs designed for bioplastic films using the Box-Behnken method.

Run	Factor 1 A (SSNP) %	Factor 2 B (SBCF) %	Factor 3 C (PLA) %
1	0.5	0	100
2	0.5	0.5	99.5
3	0	0.5	100
4	1	1	99.5
5	0.5	0	99
6	0.5	1	99
7	0	0.5	99
8	0.5	1	100
9	0	0	99.5
10	1	0.5	100
11	0.5	0.5	99.5
12	1	0	99.5
13	0	1	99.5
14	0.5	0.5	99.5
15	0.5	0.5	99.5
16	0.5	0.5	99.5
17	1	0.5	99

**Table 2 polymers-15-01557-t002:** Box-Benken experimental design responses for bioplastic films.

Response 1 (WA) %	Response 2 (TS) °C
1.2027	251
1.0693	245
1.494	335
2.9277	290.833
2.0166	241.333
3.444	262.833
1.7278	280.5
0.532	261
1.5867	270.833
1.1093	285
1.5981	251.5
1.934	242.333
2.7894	241
1.2452	238.667
1.4921	241.333
1.3711	263
1.9356	271.167

**Table 3 polymers-15-01557-t003:** ANOVA for water absorption and thermal stability of bioplastic films.

Source	Sum of Squares	dF	Mean Squares	F-Value	*p*-Values	R^2^
Water absorption (model)	7.34	9	0.8150	4.15	0.0369	0.8422
Thermal stability (model)	8254.06	9	917.12	3.85	0.0447	0.8318

Where: R2: coefficient of determination, F-value: Fisher-snedecor distribution value, df: degree of freedom, and *p*-value: probability value.

**Table 4 polymers-15-01557-t004:** Coefficients of estimates with standard errors for the model.

Factor	Water Absorption CE	Water Absorption SE	Thermal Stability CE	Thermal Stability SE
Intercept	1.36	0.1981	247.90	6.91
A-SSNP	0.0386	0.1566	−4.75	5.46
B-SBCF	0.3691	0.1566	6.27	5.46
C-PLA	−0.5983	0.1566	9.52	5.46
AB	−0.0523	0.2215	19.58	7.72
AC	−0.1481	0.2215	−10.17	7.72
BC	−0.5245	0.2215	−2.88	7.72
A²	0.3611	0.2159	26.11	7.53
B²	0.5932	0.2159	−12.76	7.53
C²	−0.1496	0.2159	18.90	7.53

**Table 5 polymers-15-01557-t005:** Optimum loading variables for water absorption and thermal stability of bioplastic films.

Independent Variables	Predicted Optimum Levels (%)
Polylactic acid	99.815
Sugar bagasse cellulose fibers	0.036
Snail shell nanoparticle	0.634

**Table 6 polymers-15-01557-t006:** Predicted and observed values for water absorption and thermal stability of bioplastic films.

Responses	Predicted Values	Observed Values
Water absorption % (model)	1.42	0.45
Thermal stability °C (model)	240.248	259.07

## Data Availability

Not applicable.

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
