# Peer review of "Assessment and Optimization of Thermal Stability and Water Absorption of Loading Snail Shell Nanoparticles and Sugarcane Bagasse Cellulose Fibers on Polylactic Acid Bioplastic Films"

_polymers, 2023, doi:10.3390/polym15061557_

Round 1
Reviewer 1 Report
The target of the present manuscript is focused on the fabrication of composites based on PLA as polymer matrix and reinforced derived from snail shell and cellulose-derived fibres to promote an improvement in water uptake resistance and thermal stability.
This topic is often studied due to the current worry of the society about sustainability of materials and the fabrication of materials friendlier with the environment. But the incorporation of the study of the design of the experiments, together with the statistical study of the results makes some interest to be considered publishable in Polymers journal. Due to the important remarks that are listed below, I suggest a rewriting process of the document and a new submission prior consideration.
- In 2.1.1. section, all the reactants to perform the conditioning of the fibres must include purity and provider, including the country too.
- In terms of components of the composite, the authors work with fibres treated to obtain nanocrystals (acidic treatment of cellulose to reduce the amorphous phase), but until page number 4 they do not mention this character. Everywhere they consider them as fibres, but they should include in the abstract the term “nanocrystal” because they must guide the reader to a specific frame of work of materials in the nanoscale. So, I encourage the authors to redefine this term in the full text.
- In section 2.1.4. the authors describe the methodology to perform the study of water uptake by immersion of samples in water and weight them after a period of immersion. They inform about the weighting of the samples, prior immersion, before the drying of the samples, and this weight is considered as “T1”, and this value is later used in the corresponding equation. But this procedure is not correct. The samples should be weighted after drying, and immediately immersed, to consider their weight at a 0 immersion time. If not, if the samples are weighed before drying, and later immersed, the naturally adsorbed water, due to the environmental moisture, evaporated during the drying process, will modify the T1 value. I suggest clarifying this methodology.
- The study has two dependent/response variables, water uptake and thermal stability. The authors perform a thermal characterization to determine the thermal stability, but they do not define which experiment they perform. They talk about the use a DSC/TGA analyzer, but they do to inform about if they perform a DSC or a TGA analysis. If the reader has enough experience in thermal analysis maybe is capable to determine that they have performed a TGA analysis, but even in this situation the authors do not inform which temperature is considered as the variable: the maximum temperature before observing a loss of weight?; the middle point between the initial weight and the final weight?; the onset of the function? etc…. Please, the author must clarify the experiment that they perform as well as the parameter that is measured. Moreover, why the authors perform a TGA experiment from 25ºC to 1000ºC? The component of the composite that is in the highest composition is PLA and after 300ºC is totally degraded. Please clarify.
- The authors talk about a design of experiment based on 29 experiments, but on table 1 are summarized just 17. What happens with the remaining? Are 29 or 17? Please clarify.
- ANOVA statistic experiment has been conducted to determine statistic variations. The authors should include in the methodology section the interval of confidence used: i.e. 95%.
- The authors inform about a dispersion procedure of the reinforcement in the solution of the polymer matrix prior casting at 65ºC for 1h. The boiling temperature of acetone is 56ºC. It is expected that the solvent could be evaporated during the process (during 1h as is reported). I consider that the authors should give more details about this procedure.
- The specific gravity of the reinforcements is higher than that of PLA. How the authors ensure the homogeneous dispersion of the reinforcements prior performing casting technique? The reinforcements will fall to the underside of the film creating an anisotropic material. Please clarify.
- In page 7 the authors talk about CNF reinforcement. I suppose that they talk about nanofibrillated cellulose, but they should give the full name in the text. If not, please clarify this topic.
- Figure 3 shows the FT-IR spectrum and they assign the band at 1080 cm-1 to C-H bending, but this signals is often observed between 1300-1500 cm-1. Maybe this signal must be attributed to C-O bending of the ester group of PLA. Moreover, the authors do not assign the important peak at 1748 cm-1. I suggest a review of the assignation of the different signals of this spectrum.
- The range of particle size that the authors inform in this document falls below 1 mm and figure 4 has a scale of 20 mm. It is impossible to observe the particles of reinforcement using this magnification. I suggest including a picture with higher magnification. And if possible, include EDAX analysis of the particles in the images to obtain the semiquantitative composition.
Author Response
Comments and Suggestions for Authors
The target of the present manuscript is focused on the fabrication of composites based on PLA as polymer matrix and reinforced derived from snail shell and cellulose-derived fibres to promote an improvement in water uptake resistance and thermal stability.
This topic is often studied due to the current worry of the society about sustainability of materials and the fabrication of materials friendlier with the environment. But the incorporation of the study of the design of the experiments, together with the statistical study of the results makes some interest to be considered publishable in Polymers journal. Due to the important remarks that are listed below, I suggest a rewriting process of the document and a new submission prior consideration.
- In 2.1.1. section, all the reactants to perform the conditioning of the fibres must include purity and provider, including the country too.
Response:
Reactants purity and provider, including the country is added into the manuscript.
- In terms of components of the composite, the authors work with fibres treated to obtain nanocrystals (acidic treatment of cellulose to reduce the amorphous phase), but until page number 4 they do not mention this character. Everywhere they consider them as fibres, but they should include in the abstract the term “nanocrystal” because they must guide the reader to a specific frame of work of materials in the nanoscale. So, I encourage the authors to redefine this term in the full text.
Response:
Cellulose nanocrystals (CNCs) and Cellulose nanofibers (CNFs) are in different in shape, size and composition. The sugarcane bagasse cellulose fibers (SBCF) used in this study is nanofiber because of their sizes and shape.
- In section 2.1.4. the authors describe the methodology to perform the study of water uptake by immersion of samples in water and weight them after a period of immersion. They inform about the weighting of the samples, prior immersion, before the drying of the samples, and this weight is considered as “T1”, and this value is later used in the corresponding equation. But this procedure is not correct. The samples should be weighted after drying, and immediately immersed, to consider their weight at a 0 immersion time. If not, if the samples are weighed before drying, and later immersed, the naturally adsorbed water, due to the environmental moisture, evaporated during the drying process, will modify the T1 value. I suggest clarifying this methodology.
Response:
Yes, Samples was weight immediately after drying in oven and the statement was corrected in the manuscript.
- The study has two dependent/response variables, water uptake and thermal stability. The authors perform a thermal characterization to determine the thermal stability, but they do not define which experiment they perform. They talk about the use a DSC/TGA analyzer, but they do to inform about if they perform a DSC or a TGA analysis. If the reader has enough experience in thermal analysis maybe is capable to determine that they have performed a TGA analysis, but even in this situation the authors do not inform which temperature is considered as the variable: the maximum temperature before observing a loss of weight?; the middle point between the initial weight and the final weight?; the onset of the function? etc…. Please, the author must clarify the experiment that they perform as well as the parameter that is measured. Moreover, why the authors perform a TGA experiment from 25ºC to 1000ºC? The component of the composite that is in the highest composition is PLA and after 300ºC is totally degraded. Please clarify.
Response:
TGA analysis was conducted to determine the thermal stability of the bioplastic. The combination of additives used for this study are naturally sources which degrade at 250 ºC due to their organic nature. However, the degrading temperature of snail shell used for developing bioplastic is 840 ºC. TGA experiment were set 25ºC to 1000ºC, but the thermal event obtained ends at 600ºC. TGA experiment has been corrected from 25ºC to 600ºC in the manuscript.
- The authors talk about a design of experiment based on 29 experiments, but on table 1 are summarized just 17. What happens with the remaining? Are 29 or 17? Please clarify.
Response:
It was a typographic error, now corrected to 17.
- ANOVA statistic experiment has been conducted to determine statistic variations. The authors should include in the methodology section the interval of confidence used: i.e. 95%.
Response:
The interval of confidence used 95% CI Bands and it is now included into the manuscript.
- The authors inform about a dispersion procedure of the reinforcement in the solution of the polymer matrix prior casting at 65ºC for 1h. The boiling temperature of acetone is 56ºC. It is expected that the solvent could be evaporated during the process (during 1h as is reported). I consider that the authors should give more details about this procedure.
Response:
The heat was not applied, and during the mixing process, acetone evaporated but not much. We observed that after some minutes, the solution formed a tick layer on the top, preventing the evaporation of the solvent. This sentence has been amended in the manuscript
- The specific gravity of the reinforcements is higher than that of PLA. How the authors ensure the homogeneous dispersion of the reinforcements prior performing casting technique? The reinforcements will fall to the underside of the film creating an anisotropic material. Please clarify.
Response:
Fast stirring techniques at 500 RPM help disperse reinforcement with a higher specific gravity in PLA successfully. SEM confirmed the uniform dispersion of this reinforcement in PLA.
- In page 7 the authors talk about CNF reinforcement. I suppose that they talk about nanofibrillated cellulose, but they should give the full name in the text. If not, please clarify this topic.
Response:
The CNF is change to SBCF in the manuscript.
- Figure 3 shows the FT-IR spectrum and they assign the band at 1080 cm-1 to C-H bending, but this signals is often observed between 1300-1500 cm-1. Maybe this signal must be attributed to C-O bending of the ester group of PLA. Moreover, the authors do not assign the important peak at 1748 cm-1. I suggest a review of the assignation of the different signals of this spectrum.
Response:
The peak at 1748cm-1 ascribed to strong C=O stretching from the δ-lactone compound linked to PLA is now included into the manuscript
- The range of particle size that the authors inform in this document falls below 1 mm and figure 4 has a scale of 20 mm. It is impossible to observe the particles of reinforcement using this magnification. I suggest including a picture with higher magnification. And if possible, include EDAX analysis of the particles in the images to obtain the semiquantitative composition.
Response:
Bioplastic was charging when get closer to lens due to excessive heat emitting from the lens. Due to this heat bioplastic tend to dissolve, which may cause damage to the machine and higher magnification was impossible. EDX studies was conducted in our previous study, it will be a repetition if is reported in current study.
Reviewer 2 Report
This manuscript brings some inspiring insights to obtain bio-plastic films. However, this study needs revision before acceptance.
- Please confirm whether the ideas in Introduction are smooth and the transition is natural. The reasons for the use of various raw materials are clearly explained, but the story is not good enough.
- The authors should re-organize the introduction to increase the readability and highlight the innovation of the manuscript compared with other works.
Author Response
Second reviewer
Comments and Suggestions for Authors
This manuscript brings some inspiring insights to obtain bio-plastic films. However, this study needs revision before acceptance.
- Please confirm whether the ideas in Introduction are smooth and the transition is natural. The reasons for the use of various raw materials are clearly explained, but the story is not good enough.
Response:
Based on the awareness of the devastation caused by plastic waste on the environment, a call for a naturally sourced bioplastic is needed, which will serve as an alternative to petroleum-based
- The authors should re-organize the introduction to increase the readability and highlight the innovation of the manuscript compared with other works.
Response:
More information is added to the introduction section.
Round 2
Reviewer 1 Report
The authors have changed the acronym CNF to SBCF but still there is a place were CNF remains. Is in table 1. Please remove it to avoid misunderstandings.
Author Response
The authors have changed the acronym CNF to SBCF but still there is a place were CNF remains. Is in table 1. Please remove it to avoid misunderstandings.
Response:
Thanks for your observation. It is corrected in the manuscript.
Regards